# Critical review of multimorbidity outcome measures suitable for low-income and middle-income country settings: perspectives from the Global Alliance for Chronic Diseases (GACD) researchers

John R Hurst ![ORCID],[1] Gina Agarwal ![ORCID],[2] Job F M van Boven ![ORCID],[3] Meena Daivadanam,[4,5,6,7,8] Gillian Sandra Gould ![ORCID],[9,10] Erick Wan-Chun Huang,[11,12] Pallab K Maulik,[13,14,15] J Jaime Miranda ![ORCID],[16] M O Owolabi ![ORCID],[17] Shahirose Sadrudin Premji,[18] Joan B Soriano,[19,20,21] Rajesh Vedanthan,[22] Lijing Yan,[23] Naomi Levitt,[24] on behalf of the GACD Multi-Morbidity Working Group

For numbered affiliations see end of article.

**Correspondence to**
Professor John R Hurst;
j.hurst@ucl.ac.uk

## ABSTRACT

**Objectives** There is growing recognition around the importance of multimorbidity in low-income and middle-income country (LMIC) settings, and specifically the need for pragmatic intervention studies to reduce the risk of developing multimorbidity, and of mitigating the complications and progression of multimorbidity in LMICs. One of many challenges in completing such research has been the selection of appropriate outcomes measures. A 2018 Delphi exercise to develop a core-outcome set for multimorbidity research did not specifically address the challenges of multimorbidity in LMICs where the global burden is greatest, patterns of disease often differ and health systems are frequently fragmented. We, therefore, aimed to summarise and critically review outcome measures suitable for studies investigating mitigation of multimorbidity in LMIC settings.

**Setting** LMIC.

**Participants** People with multimorbidity.

**Outcome measures** Identification of all outcome measures.

**Results** We present a critical review of outcome measures across eight domains: mortality, quality of life, function, health economics, healthcare access and utilisation, treatment burden, measures of 'Healthy Living' and self-efficacy and social functioning.

**Conclusions** Studies in multimorbidity are necessarily diverse and thus different outcome measures will be appropriate for different study designs. Presenting the diversity of outcome measures across domains should provide a useful summary for researchers, encourage the use of multiple domains in multimorbidity research, and provoke debate and progress in the field.

## INTRODUCTION

There is growing recognition around the importance of multimorbidity in low-income

### Strengths and limitations of this study

► There is no existing review of outcome measures suitable for use in studies to mitigate multimorbidity in low-income and middle-income country (LMIC) settings.
► The article is the written by the Global Alliance for Chronic Diseases researchers.
► It is not a systematic review.
► Further work is required to develop a core-outcome set for use in LMIC.

and middle-income countries (LMICs).[1] Multimorbidity, as defined by the UK Academy of Medical Sciences (AMS), refers to 'the coexistence of two or more chronic conditions, each of which is either a physical non-communicable disease (NCD) of long duration, a mental health condition of long duration or an infectious disease of long duration'.[1] The AMS report highlights the challenges in delivering multimorbidity research,[2] including the selection of appropriate outcome measures. In 2018, Smith completed a Delphi exercise to develop a core-outcomes set for multimorbidity research (COSmm).[3] The highest scoring outcomes were health-related quality of life, mental health outcomes and mortality. While ground-breaking, this process did not specifically target the challenges of multimorbidity in LMICs where the global burden is greatest, patterns of disease often differ and health systems are frequently fragmented.

The Global Alliance for Chronic Diseases (GACD) is an alliance of health research funders, whose research teams form a network of multidisciplinary researchers from both LMICs and high-income countries (HICs). We aim to reduce the impact of NCDs through a focus on implementation science research in LMICs, and high-priority populations in HICs. Recognising synergies across our disease-specific programmes, in 2017, we formed a Multimorbidity Working Group and published a GACD Researchers' Statement concluding that 'a greater focus on multimorbidity is overdue and necessary to successfully improve global health outcomes', thus acknowledging the specific challenge of multimorbidity in the LMIC context.[4] The statement went on to propose three strategic objectives, one of which was to change the way research is commissioned, funded and delivered when considering NCDs in LMICs.

Discussion with research funders subsequently highlighted that one barrier to funding research addressing multimorbidity in LMICs was a perceived lack of robust outcome measures. We have, therefore, developed this GACD Researchers' perspective on outcome measures suitable for studies of multimorbidity in LMICs, taking into account the challenges of (routine) data collection and patient–provider factors such as differences in interpreting social constructs and health literacy. The intent is to build on the COSmm work.[3] Derived from a common base of expertise in NCD implementation research in LMICs, we present a diversity of potential measures that can accommodate different aspects of impact in LMICs, ranging from individual-level outcomes to health service and health system effects. This is not an attempt to provide a core outcome measures set. Rather, together, the potential outcome measures inform different evaluations of effectiveness and/or process for multimorbidity. We present these as a useful resource for those designing and reviewing intervention studies for multimorbidity in LMIC settings, and hope this initiative may promote harmonisation across studies that will be essential to better map the impact of multimorbidity in LMIC settings.

## METHOD
Potential outcome measures suitable for studies of multimorbidity in LMICs were collected through a survey among the GACD multimorbidity working group, and distilled by the writing committee (the Authors) into categories through consensus discussion. All GACD researchers were invited to take part in the multimorbidity working group and those expressing interest were then invited to provide suggestions for suitable outcome measures via free-text e-mail to the group leads. In total, 31 group members participated (listed as the Authors and Contributors), with representation from all WHO Regions except the Eastern Mediterranean. GACD researchers have considerable collective experience conducting implementation science trials in LMIC settings. All measures had to be suitable for use in multimorbidity intervention studies in

LMIC, either at the individual or the population level, and from an implementation science perspective. Criteria for suitability included ease of measurement (such as availability of data, ease of data collection, availability of local translations and cost), generalisability (applicability of the proposed outcome across diverse populations within and between LMIC settings) and statistical considerations (the feasibility of demonstrating a clinically significant change with conventional statistical significance). Each outcome approach is fully described below. The initial synthesis was reviewed by members of the GACD Multimorbidity Working Group for additional comments and suggestions (the Contributors). The resulting narrative review summarises the group's collective thoughts within each domain of outcome measures studied.

### Patient and public involvement
Patients or the public were not involved in the design, conduct, reporting or dissemination plans of our research.

## OUTCOME MEASURES FOR MULTIMORBIDITY INTERVENTIONS IN LMIC
### Mortality
Death is the final common outcome for all individuals. Thus (premature) mortality is the most broadly applicable, generalisable and comparable outcome for multimorbidity research. Indeed, mortality was considered as an 'essential' core outcome measure for multimorbidity research according to the COSmm consensus.[3]

However, precisely because mortality is so broadly applicable, it suffers from a lack of specificity. While cause-specific mortality is a potential solution to the issue of specificity, this approach moves away from the goal of multimorbidity-based outcome consideration. In addition, mortality does not reflect the quality of life that an individual experiences during the time of survival; particularly in the context of multimorbidity, both disability and quality of life considerations are important in terms of an individual's experience of illness, wellness and life. Indeed, death is not always the most important outcome from a patient-centred perspective, as has been demonstrated in studies assessing patient preferences of different potential health outcomes[5–7] and conceptualised as disability-adjusted life years.

Practical challenges with mortality as an outcome measure include statistical power and sample size for an outcome that is relatively rare compared with other outcomes and proxies, potentially requiring much longer follow-up periods, except for older and/or more severely affected populations. It is, however, generally easy to measure and while the primary cause may be ascertained through techniques such as verbal autopsy (2016 WHO VA standard),[8] assessing the contribution of multimorbidity at verbal autopsy is more challenging. While misclassifying the cause of death can impact the effect size for cause-specific mortality, power will be preserved for all-cause mortality. In some LMICs, ascertainment of

deaths remains difficult due to the lack of mature vital registry systems and cultural traditions promoting deaths at home with delay in reporting.

Thus, mortality as an outcome for multimorbidity research has been infrequently used, particularly in the context of LMIC settings.[9–11] Demographic surveillance sites that have a long record of verbal autopsy could, however, provide a useful data reservoir to examine associations between multimorbidity and mortality.

## Generic quality of life scales

Health-related quality of life (HRQoL) instruments measure multidimensional well-being and functioning. Such scales may be generic such as EQ-5D and 36-item Short Form Survey (SF-36), or disease (/area) specific. While disease-specific measures may have better content and face validity as well as better responsiveness and sensitivity to change compared with generic measures, generic measures are (by definition) not disease specific and likely better for comparison of HRQoL among different diseases and for diseases in combination, an important consideration for multimorbidity research. Tools to assess the related construct of self-reported well-being have been reviewed and summarised elsewhere.[12]

Among generic tools, the COSmm consensus[3] ranked the EQ-5D, SF-36 and 12-item Short Form Survey (SF-12), and Global quality of life (WHOQOL-BREF) most highly.

The EQ-5D[13] has been widely used since introduction in the 1990s, facilitating health economic analysis (see below). It is designed to be completed by the participant and is available in multiple languages and thus widely applicable. The EQ-5D questionnaire has two components (health state description and evaluation). In the health state description, health status is measured across five dimensions: mobility, self-care, usual activities, pain/discomfort and anxiety/depression. In the evaluation section, the respondents evaluate their overall health status using a Visual Analogue Scale.

The SF-36[14] has 36 questions across eight domains: vitality, physical functioning, bodily pain, general health perceptions, physical role functioning, emotional role functioning, social role functioning and mental health.

The WHOQOL-BREF[15] is an abbreviated version of the WHOQOL-100 quality of life assessment, originally developed by the WHOQOL Group working across fifteen international field centres to develop a quality of life assessment applicable across multiple settings.

HRQoL tools have a number of advantages over mortality as an outcome, being amenable to changes in the short term. HRQoL outcomes are particularly meaningful as the aim of clinical treatment and management is generally optimising quality of life. Consequently, managing multimorbidity needs to take quality of life into account both as an outcome marker, but also an input factor into formulating clinical management. Practical considerations in LMIC include the availability of valid translations in local languages (these are more often available for the more commonly used tools, in the more commonly used languages, but coverage remains incomplete), and the challenges of use in populations with low literacy or understanding of Visual Analogue Scales. Other unanswered questions include whether thresholds for minimum clinically important difference (MCIDs) on these scales should be altered in the context of multimorbidity. Notably, some common NCDs such as hypertension are not generally associated with significant symptom burden.

## Multidimensional indices of function

The AMS[1] recommended that reports of multimorbidity should provide details of functional deficits or disabilities and frailty. In both instances, the recommendation was made that this should be coded using a standardised classification scheme. For the former, the WHO Disability Assessment Schedule 2.0 (WHODAS 2.0) or the International Classification of Functioning, Disability and Health were suggested. For the latter, the cumulative deficit model of frailty or Fried's phenotype model was recommended (see below).

### WHO Disability Assessment Schedule

The WHODAS 2.0 has been widely used in epidemiological and observational studies in LMICs. It is a self-administered 12-item questionnaire that assesses six different adult life tasks over the preceding month. The specific areas covered are: (1) understanding and communication; (2) self-care; (3) mobility; (4) interpersonal relationships; (5) work and household roles and (6) community and civic roles. WHODAS has been included as a secondary outcome measure in three multimorbidity trials in LMIC (currently unreported[16 17]).

### Frailty assessment instruments

There are many methods to assess frailty including the Fried Index, the Frailty Index and the British Frailty Index. While these have been used to examine the prevalence, correlates or outcomes of frailty in LMIC, further validation is still required in these settings.[18] Of the various metrics, the Fried Index[19] has been the most commonly used in LMIC. This index measures frailty by the presence of three or more of five physical deficits—exhaustion, weakness, slowness, low levels of activity and weight loss. Three of the items are collected using questionnaires, but slowness is assessed using a walking test and weakness by assessing grip strength. The Frailty Index has also been commonly used in LMICs and uses the presence or absence of medical conditions or poor performance on functional tasks to assess the number of deficits present and thus frailty.[18] Using frailty as an outcome measure for intervention studies in patients with multimorbidity in LMIC may be limited by factors such as a lack of equipment (eg, to measure grip strength), and the question remains as to how susceptible to change such measurements are, and what an MCID might be. Despite this, frailty instruments remain an important outcome in LMIC settings given that frailty may be a confounding

factor in self-care, treatment adherence and family burden.

## Assessment of physical functioning

Physical functioning measures are commonly studied outcomes. The most frequently used indices include activities of daily living (ADL) (such as eating, dressing and toileting), instrumental activities of daily living (such as shopping and answering phone calls) and the Barthel Index (self-reported outcomes on degree of assistance needed for mobility, self-care and continence). Smith *et al*[8] described ADL, physical function and physical activity as core outcomes in multimorbidity interventions. For ADL the following measures received greatest support: Frenchay Activities Index, Nottingham Extended Activities of Daily Living and the Instructions for Activities of Daily Living questionnair, but these have not been evaluated in the context of LMICs.

The modified Rankin Scale is an example of a disease-specific (in this case, stroke) composite outcome measure including rating of functioning from no interference with daily life, through various degrees of disability to death. These outcomes are relatively easy to assess and have particular relevance in LMICs as people generally express strong desires in maintaining physical functioning including their ability to work, avoiding financial consequences and burden on family caregiving.

## Health economic implications

The AMS report[1] highlighted the economic burden of multimorbidity in LMICs and thus health economic implications are relevant in any consideration of multimorbidity outcome measures. However, most economic data on multimorbidity were gathered in HICs and the question arises as to whether measurement instruments, data and outcomes commonly used to assess cost implications of multimorbidity in HICs are applicable to LMIC settings.

One of the most common economic evaluations of healthcare interventions makes use of a technique called cost-effectiveness analysis and specifically the incremental cost-effectiveness ratio (ICER).[20] The method to calculate the ICER is not disease specific, making it just as suitable to assess multimorbidity interventions as single disease interventions. However, it requires specific attention to the definitions and collection of costs and effect data in LMICs. Within this ratio, costs and effects can be defined, measured and calculated in different ways, of which some are more suitable in economic assessment of multimorbidity interventions in LMICs than others. Interpretation of the ratio may differ in different settings.

In healthcare, interventions can impact different types of direct and indirect costs within and outside healthcare systems. The different costs to be included in cost-effectiveness analysis depends on the perspective that is taken (eg, the healthcare payer, the society, the patient or the family). Costs that directly result from the intervention and which occur within healthcare systems should be included when a healthcare payer perspective is taken. However, in LMICs that lack universal health coverage, the perspective of the patient and family may be more relevant and a key focus could be on out-of-pocket costs. Examples of indirect costs are work productivity losses and these costs are especially relevant when a patient or societal perspective is taken.

In health economic studies, the effect of intervention uses a measure that is independent of a specific disease: the quality-adjusted life year (QALY). The QALY is a combination of utility (preferably measured using the EQ-5D) and survival. With the EQ-5D, certain health states are defined, to which a specific utility is assigned. Utility is the value a society gives to a specified health state and for each country a specific algorithm should be estimated from large general population samples. In many LMIC settings, these still need to be further developed to allow for generalisable models of effectiveness.

While most HICs have defined guidelines and make use of fixed thresholds or ranges to assess whether a certain ICER is considered cost-effective, such guidelines and thresholds are generally lacking in LMICs. This complicates the interpretation of cost-effectiveness analyses in LMICs. As a general rule, WHO defines an intervention that costs less than three times the gross domestic product per capita as cost-effective.[21] It is important to note that the implications of economic analyses discussed here are not challenges specific to multimorbidity, but are nonetheless suitable for the study of multimorbidity.

## Healthcare access and utilisation

Multimorbidity is associated with repeated care seeking, often at different providers. This not only results in multiple interactions with healthcare settings through outpatient and inpatient admissions but also involves paramedical services and practitioners of traditional medicine.

Although we identified no study that has specifically looked at generating or testing multimorbidity related healthcare access indices in LMICs, the WHO Study on Global Ageing and Adult Health which focused on LMICs tracked indicators specific to multimorbidity in ageing populations.[22 23] These included the number of outpatient visits in the last 12 months, overnight hospital stays in the past 3 years, and the number of overnight stays in hospital in the past 12 months. A UK National Health Service document[24] outlines equity indicators that may also map multimorbidity relevant in LMIC settings, and some of these have direct healthcare access relevance such as emergency hospitalisations for chronic conditions and repeat emergency hospitalisations in the same year. Access to medicines listed on the WHO Essential Medications list would provide another metric, as would recommendations on attention to comorbidity and pharmacological interactions in treatment guidelines.

This lack of LMIC specific multimorbidity indices to plot healthcare access leads to a critically important avenue of research that could draw on that conducted

in HICs.[25] The latter work lists a range of objectives that need to be addressed in healthcare practices catering to clients with multimorbidity and lists a set of preventive services for such cases which health facilities should provide. Health-seeking behaviour is a further dimension related to healthcare access that is shaped by unique socioeconomic and cultural contexts faced by patients in LMICs. We suggest it would be useful to develop health-seeking behaviour indices relevant across LMICs. This needs a contextual framework to best understand what is feasible and what can be tracked within specific LMIC settings, acknowledging the challenges introduced by the fragmentation of care and the multiplicity of levels of provision of care in the public and private sectors. Such indices could be linked with existing monitoring frameworks used to assess Universal Health Coverage.[26]

The Global Burden of Disease initiative has recently incorporated a new metric at national level termed the Healthcare Access and Quality (HAQ) Index.[27] The HAQ index is a scale from 0 to 100, calculated by measuring mortality rates from causes that should not be fatal (amenable mortality) in the presence of effective medical care. This correlates with the Sociodemographic Index, a measure of overall development consisting of income per capita, average years of education and total fertility rates.

## Treatment burden

The burden of treatment, a relatively new concept, emerged from disease-centred healthcare systems in response to the growing needs of coping with chronic conditions. In the context of multimorbidity, this may be considered as the workload and impact on a patient as a result of receiving medical care.[28] High treatment burden may lead to overwhelmed patients who struggle to access healthcare and adhere to suggested treatment while coordinating their own care and other aspects of life, a particular issue among patients with multimorbidity. As a consequence, polypharmacy and non-adherence to treatment and poor clinical outcomes may follow, resulting in an even higher burden of treatment, a deterioration cycle depicted in the Cumulative Complexity Model.[29] Therefore, assessing treatment burden is a priority in order to achieve better quality healthcare, and treatment burden is a potential outcome measure in interventions directed against multimorbidity. There is also the challenge, more pronounced in LMICs, that in areas of no care there can be no 'burden' from treatment which it is impossible to access.

Assessing the burden of treatment is not an easy task. It generally requires multidimensional measures that are tailored to the medical condition(s), health system(s) and cultural background. Tailoring to specific conditions may diminish value in multimorbidity. Eton *et al* proposed a conceptual framework of treatment burden based on qualitative inquiries to patients with chronic conditions, consisting of 3 themes and 15 subthemes.[30] A number of tools for evaluating treatment burden for patients with multimorbidity have been developed in the past

few years. Eton *et al* designed and validated the Patient Experience with Treatment and Self-management.[31] The Treatment Burden Questionnaire (TBQ) is another instrument, consisting of 15 items[32] and later further adapted.[33 34] In 2018, Duncan published the Multimorbidity TBQ, a 10-item measure initially validated in primary care in the UK.[35] The Healthcare Task Difficulty questionnaire is an 11-question tool designed to measure only one aspect—perceived difficulty in performing healthcare management tasks.[36] Finally, the Multimorbidity Illness Perceptions Scale, unlike other instruments, was designed to measure the perceived impact of multimorbidity.[37] The scale includes treatment burden (six questions) as one of the subscales.

As these questionnaires are relatively new, validation and translation for different populations and geographical areas remain limited, especially in LMICs. Exploring the notion and measurement of treatment burden in LMIC remains relatively unexplored,[34 38] as does the important concept of patient-reported experience measures in LMIC settings which may themselves affect health outcomes.[39]

There are a number of remaining issues to be considered before applying these tools in LMICs. First, the strengths and limitations of each tool should be examined as careful validation has often not been conducted in such settings. Second, using mixed-methods incorporating experiences and opinions from patients and healthcare providers may help identify relevant issues relating to differences in contexts, cultures and health system structures. Third, as all of these instruments have been available for less than a decade, longitudinal evidence of change over time is absent.

## Measures of 'Healthy Living'

Multimorbidity is complex to operationalise, which makes common denominators very relevant. Measures of 'Healthy Living' are direct common denominators for being at risk of developing individual components of multimorbidity, and thus measuring change in these measures provides potential generic outcomes of interventions to mitigate future multimorbidity. Most current behavioural interventions have targeted only one behaviour at a time.

Healthy Living encompasses many different aspects of health and well-being, including diet, physical activity including sedentary behaviour, tobacco and alcohol consumption, developing health literacy, maintaining good hygiene and sanitation. Most current behavioural interventions have targeted only one behaviour at a time.

## Diet

Dietary assessments are complex. Self-reported dietary intake measurements are the most common form of dietary assessments, which include prospective recording of actual food consumed or retrospective recall.[40 41] With respect to multimorbidity, the focus must be on long-term usual intake. Dietary diversity scores are one such

measure that can be estimated for the individual, or the household using counts of food items (food variety score) or food groups (dietary diversity score) consumed over a prespecified period.[42 43] Dietary diversity can be estimated at the Household level using the Household Dietary Diversity Score, which assesses household access to a variety of foods, or at individual level for women and children respectively using the Minimum Dietary Diversity for Women of Reproductive Age tool and WHO Infant and Young Child Minimum Dietary Diversity Tool.[44]

### Physical activity including sedentary behaviour

Convincing interventional evidence showing a clear dose–response relationship between physical activity (PA) and improved health outcomes comes mainly from HICs, although associations of PA with reduced cardiovascular mortality and morbidity are available globally.[45] Sedentary behaviour, defined as those that involve sitting or reclining and low levels of energy expenditure during waking hours,[46] has also been associated with having at least two morbidities, independent of light, moderate or vigorous PA[46 47] in HICs and LMICs. The Global Physical Activity Questionnaire (GPAQ) that is part of the WHO STEPwise Approach to Chronic Disease Risk Factor Surveillance data collection tool[48] is a commonly used tool to collect self-reported data on PA. The GPAQ which is a shorter (16-item) version of the longer International Physical Activity Questionnaire also assesses sitting time in addition to PA in three domains (work, travel and leisure time). This is used to estimate the duration of moderate to vigorous PA or intensity in terms of metabolic equivalent minutes per week of total and domain-specific activities. However, agreement between PA estimated by GPAQ and more objective measures has been moderate at best. Objective measures of PA allow real-time monitoring and can be easily completed using an application on a mobile device or a wearable pedometer or accelerometer, although this has mostly been tested in HIC settings. Considering the rapid acceleration of smart phone ownership in LMIC, and the availability of cheaper but robust wearable devices, these are now viable options and an optional tool to capture objective PA has since been incorporated into the GPAQ.

### Tobacco and alcohol use

Tobacco use has been consistently linked as a causative factor for chronic respiratory disorders such as chronic obstructive pulmonary disease, cardiovascular disease and many cancers including lung cancer. Similarly, alcohol use has strong associations with NCDs. Ever and current use of tobacco or current use of alcohol are commonly used assessments in addition to questions focusing on frequency and amount of consumption, and these are part of the WHO STEPS instrument.[48] Where available, verification of smoking status can be achieved through measurement of carbon monoxide or urinary cotinine. Assessment of household, environmental and occupational airborne exposures are more complex.

### Healthy Living Index

In addition to individual risks and behaviours, composite indicators that assess Healthy Living may be more relevant in the context of multimorbidity. Tools to assess the environment in terms of its potential to offer opportunities for Healthy Living have been limited, especially in LMICs. Environmental Profile of a Community's Health (EPOCH) is a quantitative tool designed to capture community perceptions of tobacco, nutrition and social environments, validated in five countries (China, India, Brazil, Colombia and Canada).[49 50] EPOCH comprises an objective assessment of the physical environment, and an interviewer-administered questionnaire on residents' perceptions of their community to capture both objective and subjective measures of the environment.[49] The Community Healthy Living Index developed in the USA assessed the environmental support potential of a community across five domains assessing a specific venue: schools, afterschool child care sites, work sites, neighbourhoods and communities-at-large.[51] Such tools could be adapted for use in LMICs.

### Self-efficacy and social functioning

Self-efficacy and social functioning relate to social determinants of health such as age, gender, marital status, family background, employment, education level and socioeconomic status,[52–58] affecting in turn how an individual is able to look after their health conditions (self-efficacy) and interact in society with other individuals leading a fulfilling life (social functioning). This raises the important question of whether indices of self-efficacy and social functioning may be suitable as outcomes measures in studies to mitigate multimorbidity in LMIC settings.

There are limited studies that explore which social determinants are more influential than others in determining self-efficacy and social functioning. Positive personality traits and higher self-esteem demonstrated in adolescence positively affect self-efficacy.[59] Competent behaviour, such as skills of focusing on others' well-being, affiliative behaviours/interpersonal cooperation and participation, which are culturally valued and socially competent are associated with higher self-efficacy.[60 61] Liebke et al[62] reported that loneliness and social functioning are associated. Loneliness may be caused by impaired social skills, such as maintaining conversations or expressing feelings, which are essential to adequate social functioning.[62] Values placed on social determinants of health may vary across different cultures. Differences in cultural traditions may affect the sources of self-efficacy belief systems.[52 55 56]

Given the multitude of cultural factors affecting the precursors of self-efficacy and social functioning, populations in LMICs may have fewer opportunities to develop such skills. Therefore, while measures of self-efficacy and social functioning could be used as multimorbidity outcome measure in LMIC, a single index is unlikely to be useful across all settings.

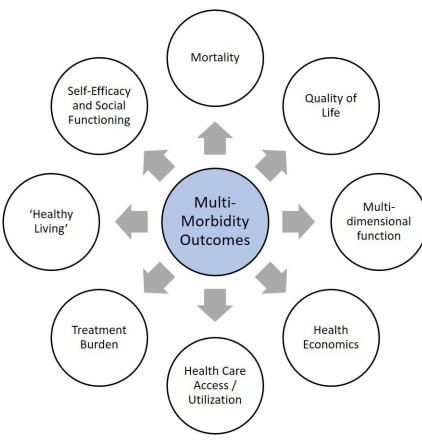

**Figure 1** Eight domains of outcome measures for multimorbidity interventions in LMIC. LMIC, low-income and middle-income country.

## CONCLUSIONS

The case has been made for the growing global importance of multimorbidity, the need for pragmatic intervention studies to reduce the risk of developing multimorbidity in LMIC settings, and of mitigating the complications and progression of multimorbidity. One of many challenges in such research has been the selection of appropriate outcomes measures.

We present the GACD Researchers' perspective on outcome measures suitable for multimorbidity intervention studies in the context of LMICs. We have considered outcome measures across eight domains (figure 1). Some represent direct measurements of clinical outcomes, while others represent intermediate variables on the pathway to multimorbidity. Some measures are single, others are composite. They vary in their ease of collection and cost. It is critical to choose appropriate outcomes for the study design, cultural context and participant preference in order to demonstrate and understand the effect of an intervention, and our aim is therefore not to suggest a preference of one outcome measure over any other. Studies in multimorbidity are necessarily diverse and thus different outcome measures will be appropriate for different study designs. As with the COSmm consensus,[3] we recognise the key importance of mortality and HRQoL as multimorbidity outcomes, and these are suitable for use in LMIC settings. Many other outcomes from the COSmm work, including patient-reported impacts and behaviours (such as treatment burden and self-efficacy), physical activity and function, and health systems indicators (notably health economic indices) are also suitable for LMIC settings, though in the context and with the caveats that we have described above. Some of the challenges applying these outcome measures in LMIC are also relevant in HIC.

The diversity of outcome measures across domains demonstrated here should provide a useful summary for researchers, and encourage the use of multiple domains in multimorbidity research, rather than just a single outcome measure. Ultimately, the proof of utility for these outcome measures will be the demonstration that an effective multimorbidity intervention can improve the health of the community in which it is tested. Meanwhile, there remains the urgent need for further study and development of outcome measures suitable for multimorbidity intervention studies in the context of LMIC.

There are limitations to this work, which is not intended to be a core outcome set, nor a systematic review. Development of both these would be an important contribution to the field, as would further work to understand the perceptions of these outcome measures from people directly affected by multimorbidity and tools suitable for assessing patient-reported experience in the context of multimorbidity. Here, we present a critical, narrative synthesis describing the range of outcome measures that might be selected for use in such settings, and their challenges. The key strength of our work is the broad representation of views from GACD researchers who have considerable collective experience of implementation science research in LMIC settings. We anticipate this will be useful to other researchers designing and conducting such studies, and to provoke debate and progress in the field.

**Author affiliations**
[1]UCL Respiratory, University College London, London, UK
[2]Family Medicine, McMaster University, Hamilton, Ontario, Canada
[3]Department of Clinical Pharmacy & Pharmacology, University Medical Center Groningen, Groningen, The Netherlands
[4]Deptartment of Global Public Health, Karolinska Institutet, Stockholm, Sweden
[5]Department of Food, Nutrition and Dietetics, Uppsala University, Uppsala, Sweden
[6]Department of Food Studies, Nutrition and Dietetics, Uppsala University, Uppsala, Sweden
[7]Deptartment of Global Public Health, Karolinska Institutet, Solna, Sweden
[8]International Maternal and Child Health Division, Department of Women's and Children's Health, Uppsala University, Uppsala, Sweden
[9]School of Medicine and Public Health, The University of Newcastle, Callaghan, New South Wales, Australia
[10]Hunter Medical Research Institute, New Lambton, New South Wales, Australia
[11]Respiratory and Environmental Epidemiology, Woolcock Institute of Medical Research, Sydney, New South Wales, Australia
[12]Division of Thoracic Medicine, Department of Internal Medicine, Taipei Medical University Shuang Ho Hospital, New Taipei City, Taiwan
[13]Research, The George Institute for Global Health, New Delhi, India
[14]Faculty of Medicine, University of New South Wales, Sydney, New South Wales, Australia
[15]Prasanna School of Public Health, Manipal, India
[16]CRONICAS Centre of Excellence in Chronic Diseases, Universidad Peruana Cayetano Heredia, Lima, Peru
[17]Medicine, University of Ibadan College of Medicine, Ibadan, Nigeria
[18]School of Nursing, Faculty of Health, York University, Toronto, Ontario, Canada
[19]Universidad Autónoma de Madrid, Madrid, Spain
[20]Hospital Universitario de la Princesa, Instituto de Investigación Sanitaria Princesa (IP), Madrid, Spain
[21]Centro de Investigación en Red de Enfermedades Respiratorias (CIBERES), Instituto de Salud Carlos III (ISCIII), Madrid, Spain
[22]Department of Population Health and Department of Medicine, NYU Langone Health, New York, New York, USA
[23]Global Health Research Center, Duke Kunshan University, Jiangsu, China
[24]Medicine, University of Cape Town, Cape town, South Africa

**Correction notice** This article has been corrected since it was published. The licence has been updated.

**Collaborators** GACD Multimorbidity Working Group: Ricardo Araya, Kirsten Bobrow, Niels H Chavannes, F Xavier Gómez-Olivé, Shabbar Jaffar, Bruce J Kirenga, Rianne M J J van der Kleij, Muralidhar M Kulkarni, Laura Loli-Dano, Patricio Lopez-Jaramillo, Shane Norris, Josefien van Olmen, Gary Parker, Trishul Siddharthan, Kamran Siddiqi, Najma Siddiqi, Antigona C Trofor.

**Contributors** The concept for this paper arose from discussion at the GACD Multimorbidity Working Group chaired by JRH. The authors (JRH, GA, JFMvB, MD, GSG, EW-CH, PKM, JJM, MOO, SP, JS, RV, LY and NL) planned and conducted the e-mail survey. All authors and contributors (RA, KB, NHC, FXG-O, SJ, BJK, RMJJvdK, MMK, LL-D, PL-J, SN, JvO, GP, TS, KS, NS and ACT) provided suggestions for outcome measures. Individual sections of the manuscript were drafted by the authors, coordinated by JRH into a first complete draft. All authors revised this initial draft. All authors and contributors provided important intellectual content on the revised draft and approved the final version for submission.

**Funding** The article was supported by the Global Alliance for Chronic Diseases (GACD) secretariat. The Authors are investigators on individual studies funded in collaboration with the GACD.

**Competing interests** None declared.

**Patient and public involvement** Patients and/or the public were not involved in the design, or conduct, or reporting, or dissemination plans of this research.

**Patient consent for publication** Not required.

**Provenance and peer review** Not commissioned; externally peer reviewed.

**Data availability statement** Data sharing not applicable as no datasets generated and/or analysed for this study.

**ORCID iDs**
John R Hurst http://orcid.org/0000-0002-7246-6040
Gina Agarwal http://orcid.org/0000-0002-5691-4675
Job F M van Boven http://orcid.org/0000-0003-2368-2262
Gillian Sandra Gould http://orcid.org/0000-0001-8489-2576
J Jaime Miranda http://orcid.org/0000-0002-4738-5468
M O Owolabi http://orcid.org/0000-0003-1146-3070

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
