## [Reviewer comments · BMJ Open]

ARTICLE DETAILS

TITLE (PROVISIONAL)	A Critical Review of Multi-Morbidity Outcome Measures suitable for Low- and Middle Income Country Settings: perspectives from the Global Alliance for Chronic Diseases (GACD) Researchers
AUTHORS	Hurst, John; Agarwal, Gina; van Boven, Job F. M.; Daivadanam, Meena; Gould, Gillian; Wan-Chun Huang, Erick; Maulik, Pallab; Miranda, J. Jaime; Owolabi, M. O.; Premji, Shahirose; Soriano, Joan; Vedanthan, Rajesh; Yan, Lijing; Levitt, Naomi

VERSION 1 – REVIEW

REVIEWER	Sarah Gorst University of Liverpool, UK
REVIEW RETURNED	17-Feb-2020

GENERAL COMMENTS	The authors have provided a critical review of outcome measures for studies of multi-morbidity in LMIC settings. The authors are clear from the outset that they are not reporting a systematic review and are not developing a core outcome set. Instead, they are summarising and critically reviewing outcome measures for this specific population. However, I think that it would have been more appropriate to build on the COSmm work to ascertain what outcomes should be measured in all multi-morbidity studies in LMIC settings and then moving onto reviewing the outcome measurement instruments that could be used to measure these specific outcomes. Nevertheless, due to the complexity of multi-morbidity studies and the diversity of outcome measures used, I think this review will be of use to researchers working in this field. I just have a couple of minor comments that I would like that authors to address. 1. I would have liked to see more detail in the methods section of the manuscript. Specifically, to include more detail on the survey used to collect outcome measures among the GACD multi-morbidity working group and how the authors organised these outcome measures into categories.2. There is no discussion about the study limitations in the main text of the manuscript. There are a couple of points listed after the abstract, but no detail provided. Therefore, I suggest that the authors add some detail about the limitations into the conclusion section of the manuscript.3. There are some instances where full stops are missing from the end of sentences and so I would urge the authors to correct these errors.
--

REVIEWER	Maureen Rutten-van Mólken Erasmus University Rotterdam
REVIEW RETURNED	21-Feb-2020

GENERAL COMMENTS	Outcome measures suitable for use in persons with multimorbidity (MM) from LMIC is an important topic as these countries generally have less resources to conduct comprehensive research and to routinely monitor outcomes. The objective of this paper is to summarize and review outcome measures suitable for MM research in LMIC. How 'suitable for use in LMIC' was defined needs further explanation. Details about the methods of this study are lacking. I have doubts whether a survey is a suitable method to summarize and review outcome measures, as respondents cannot be expected to have a complete overview of available outcome measures. Details on the background of people interviewed, the countries, the number of interviewees etc should be provided. Details about the content of the survey are not provided. A literature review seems to be done as well, but this is not mentioned in the methods and it appears not to be done systematic. What was the strategy to identify the questionnaires mentioned? There is a review by Linton et al., of 99 self-reported measures of well-being published in BMJ Open, which could be useful. How the working group came to the classification of outcome measures is the results section and how they judged suitability in LMIC needs further explanation. Patients were not involved in this study at all, but especially in MM, the involvement of patients with MM is important to identify what are important outcome measures. In the results section, issues related to the suitability of outcome measures in LMIC are not systematically addressed for every category of outcomes. Many of the issues that are mentioned are applicable to high income countries as well (e.g. availability of translations, validity of questionnaires, lack of a clear cost-effectiveness threshold). Measures of healthy living seem to be equally important in high income countries. Availability of translations is mentioned in the section on generic QoL scales, but especially for these scales a high number of translations is available. Info can be found on the websites of EQ-5D, SF-36, WHO. Why would frailty measures be more influenced by lack of equipment in LMIC than in high income countries? I would not treat the health economic indicators as an outcome measure, because the question to be answered is how much it costs to gain one additional unit of an outcome measure (usually a QALY). This is not more relevant in a LMIC than in a high income country. The question about which perspective to take for the economic evaluation does not effect the choice of outcome measure but the choice of costs categories to include. I was surprised that outcomes measuring a patient's experience with care were not mentioned, because especially patients with MM suffer from the fragmentation in the system. I acknowledge
---

	that this is partly included in measures of treatment burden, but there are many more experience indicators. One such indicator is 'continuity of care', which seems to be a very important outcome measure for LMIC. Time between referral and start of treatment can be long when the availability of services is limited. Why was continuity left out? For self-efficacy, variables that influence this outcome measure are discussed. It is unclear why that was added for this outcome measure in particular. In summary, I would suggest that the authors provide a much more comprehensive and systematic analyses of why some instruments would be more suited for use in LMIC than others, based on the concepts that are measured by these instruments, the practical administration of the instruments, their costs (some are not available for free), etc.
--	--

VERSION 1 – AUTHOR RESPONSE

Reviewer 1; Sarah Gorst; University of Liverpool.

R1C1. The authors have provided a critical review of outcome measures for studies of multi-morbidity in LMIC settings. The authors are clear from the outset that they are not reporting a systematic review and are not developing a core outcome set. Instead, they are summarising and critically reviewing outcome measures for this specific population. However, I think that it would have been more appropriate to build on the COSmm work to ascertain what outcomes should be measured in all multi-morbidity studies in LMIC settings and then moving on to reviewing the outcome measurement instruments that could be used to measure these specific outcomes. Nevertheless, due to the complexity of multi-morbidity studies and the diversity of outcome measures used, I think this review will be of use to researchers working in this field. I just have a couple of minor comments that I would like that authors to address.

R1R1. Thank you.

R1C2. I would have liked to see more detail in the methods section of the manuscript. Specifically, to include more detail on the survey used to collect outcome measures among the GACD multi-morbidity working group and how the authors organised these outcome measures into categories.

R1R2. Suggestions for outcome measures were invited from the GACD multi-morbidity working group by free-text email. These were organised into categories by consensus discussion among the Authors. We have added further detail to the text of the Method section on page 3 to clarify this.

R1C3. There is no discussion about the study limitations in the main text of the manuscript. There are a couple of points listed after the abstract, but no detail provided. Therefore, I suggest that the authors add some detail about the limitations into the conclusion section of the manuscript.

R1R3. Thank you for this suggestion. We have added a new paragraph summarising the limitations of this work in the Conclusion section, page 12.

R1C4. There are some instances where full stops are missing from the end of sentences and so I would urge the authors to correct these errors.

R1R4. Sorry; we have carefully proof-read the revised version.

Reviewer 2; Maureen Rutten-van Mölken; Erasmus University Rotterdam.

R2C1. Outcome measures suitable for use in persons with multimorbidity (MM) from LMIC is an important topic as these countries generally have less resources to conduct comprehensive research and to routinely monitor outcomes. The objective of this paper is to summarize and review outcome measures suitable for MM research in LMIC. How 'suitable for use in LMIC' was defined needs further explanation.

R2R1. Thank you for your comprehensive review of our paper, and constructive comments. We defined 'suitable for use in LMIC' based on ease of measurement, generalizability and statistical considerations and have added further detail on this to the Method, page 3. The GACD is a network of researchers united through a shared implementation science approach to non-communicable diseases in LMIC, and thus the group were familiar with the strengths and weaknesses of the suggested outcomes.

R2C2. Details about the methods of this study are lacking. I have doubts whether a survey is a suitable method to summarize and review outcome measures, as respondents cannot be expected to have a complete overview of available outcome measures. Details on the background of people interviewed, the countries, the number of interviewees etc should be provided. Details about the content of the survey are not provided. A literature review seems to be done as well, but this is not mentioned in the methods and it appears not to be done systematic. What was the strategy to identify the questionnaires mentioned? There is a review by Linton et al., of 99 self-reported measures of well-being published in BMJ Open, which could be useful.

R2R2. Reviewer 1 also asked for further clarification of our Method and we have provided further detail in the revised manuscript on page 3. We have clearly acknowledged that this is not a systematic review or formal literature review. The working group was surveyed by e-mail with 31 people in total contributing free-text suggestions for outcome measures (including questionnaires);

these people are listed as Authors or Contributors and their affiliations are provided in the manuscript meta-data which will be included at publication. Collectively, the group has considerable experience of delivering clinical trials in LMIC settings. We received responses from researchers based in all WHO regions, except the Eastern Mediterranean (and some of our researchers have worked in this region). Including reference to the review by Linton is a helpful suggestion, thank you, and we have added this on page 4.

R2C3. How the working group came to the classification of outcome measures in the results section and how they judged suitability in LMIC needs further explanation.

R2R3. Outcome measures were organised into categories by consensus discussion among the Authors and we have added a note on this in the Method, page 3. As described above in R2R1, we defined 'suitable for use in LMIC' based on ease of measurement, generalizability and statistical considerations and this is also described in further detail in the Method, page 3. GACD researchers have considerable collective experience delivering implementation trials in LMIC settings.

R2C4. Patients were not involved in this study at all, but especially in MM, the involvement of patients with MM is important to identify what are important outcome measures.

R2R4. We agree that understanding the views of people living with multi-morbidity is important. Our work here is a survey of potential outcomes from the perspectives of researchers. Understanding the relative merits of these outcomes from the perspectives of people affected by multi-morbidity would be important future work. We had included specific text acknowledging this in the expanded limitations section, page 12.

R2C5. In the results section, issues related to the suitability of outcome measures in LMIC are not systematically addressed for every category of outcomes. Many of the issues that are mentioned are applicable to high income countries as well (e.g. availability of translations, validity of questionnaires, lack of a clear cost-effectiveness threshold). Measures of healthy living seem to be equally important in high income countries.

R2R5. We agree that many of the challenges described are also relevant in HIC, addressing that is not the purpose of our Review but we have included a note to this effect in the Conclusion section (page 12). We agree also that measures of healthy living are also important in HIC. We had included commentary on suitability for use in LMIC across each of the eight outcome measure domains.

R2C6. Availability of translations is mentioned in the section on generic QoL scales, but especially for these scales a high number of translations is available. Info can be found on the websites of EQ-5D, SF-36, WHO.

R2R6. Thank you, we are aware of these, and for the EQ-5D this is specifically acknowledged on page 4 (“available in multiple languages”). However, the point remains that there are few fully and formally validated questionnaires in local languages and dialects beyond official languages in LMIC. We have added clarifying text on page 5.

R2C7. Why would frailty measures be more influenced by lack of equipment in LMIC than in high income countries?

R2R7. The need for equipment to measure, for example, grip strength was identified as a potential barrier to the use of some frailty tools in LMIC settings, and we included an explanation of that on page 5.

R2C8. I would not treat the health economic indicators as an outcome measure, because the question to be answered is how much it costs to gain one additional unit of an outcome measure (usually a QALY). This is not more relevant in a LMIC than in a high income country. The question about which perspective to take for the economic evaluation does not effect the choice of outcome measure but the choice of costs categories to include.

R2R8. There are multiple perspectives on this issue but we agree that we should have used clearer terminology. We have therefore adjusted the first and last sentence and the subheading to indicate, in agreement with the Reviewer, that cost-effectiveness itself may not be a direct outcome measure. Nevertheless, in LMIC settings, there are specific challenges when economic implications of multi-morbidity outcomes are measured and we feel that these should be discussed. At a population level, economic indicators can be an effective way to assess the impact of complex interventions on people living with or at risk of multi-morbidity.

R2C9. I was surprised that outcomes measuring a patient's experience with care were not mentioned, because especially patients with MM suffer from the fragmentation in the system. I acknowledge that this is partly included in measures of treatment burden, but there are many more experience indicators.

R2R9. We agree that patient perspectives are important (see response to R2C4 above). We do feel that there is an important distinction between health outcome, on the one hand, and experience of health care on the other, and our review is only designed to assess the former. We agree that patient experience of care is important and have included new text on that in the section on treatment burden (page 9). Experience of health care can impact health outcomes and we have added reference to the meta-analysis by Birkhäuser examining relationships between trust in health-care staff and health outcomes.

R2C10. One such indicator is 'continuity of care', which seems to be a very important outcome measure for LMIC. Time between referral and start of treatment can be long when the availability of services is limited. Why was continuity left out?

R2R10. Continuity of care can have multiple definitions and interpretations. We agree that for a single condition, time between initial referral and start of treatment is an important process outcome. However, we feel that this is better considered a process indicator rather than a health outcome per se.

R2C11. For self-efficacy, variables that influence this outcome measure are discussed. It is unclear why that was added for this outcome measure in particular.

R2R11. Thank you. We agree that there is more detail in this section. However, we believe many readers will be less familiar with the concept of self-efficacy compared to, for example, mortality or quality of life. Our preference is therefore to leave this text unaltered, but we would be happy to remove some of the details at the Editor's request.

R2C12. In summary, I would suggest that the authors provide a much more comprehensive and systematic analyses of why some instruments would be more suited for use in LMIC than others, based on the concepts that are measured by these instruments, the practical administration of the instruments, their costs (some are not available for free), etc.

R2R12. Thank you for this comment. Our intent is to describe the spectrum of possible outcome measures, to inform other researchers in the field, in order to help researchers make their own decision about which outcome measures to use depending on specific study characteristics, patient preference, cultural context and research question. We are not presenting some instruments as more suited than others, as the choice of outcome measure will crucially depend on these factors. We have added further text to the Conclusion section on pages 11 and 12 to clarify these points.

VERSION 2 – REVIEW

REVIEWER	Sarah Gorst University of Liverpool, UK
REVIEW RETURNED	12-Jun-2020
GENERAL COMMENTS	The authors have addressed all my comments and I am happy for this manuscript to be published in BMJ Open.